# An Investigation of Representative Customer Load Collectives in the Development of Electric Vehicle Drivetrain Durability

Mingfei Li [1], Fabian Kai-Dietrich Noering [1], Yekta Öngün [1], Michael Appelt [1,*] and Roman Henze [2]

[1] Volkswagen AG, 38436 Wolfsburg, Germany; mingfei.li1@volkswagen.de (M.L.); fabian.kai-dietrich.noering@volkswagen.de (F.K.-D.N.); yekta.oenguen@volkswagen.de (Y.Ö.)

[2] Institute of Automotive Engineering, Technische Universität Braunschweig, 38106 Braunschweig, Germany; r.henze@tu-bs.de

* Correspondence: michael.appelt@volkswagen.de; Tel.: +49-151-16753488

**Abstract:** To ensure the precise dimensioning and effective testing of drivetrain components, it is crucial to have a thorough understanding of customer requirements, with a particular emphasis on customer stress on these components. An accurate interpretation of customer data is essential for determining representative customer requirements, such as load collectives. The automobile industry has faced challenges in analyzing large amounts of customer driving data to obtain representative load collectives as target values in durability design. However, due to technical limitations and cost constraints, collecting data from a large sample size is not feasible. The ongoing digitalization of the automotive industry, driven by an increasing number of connected vehicles, enhances data-based and customer-oriented development. This paper investigates representative customer load collectives using cloud data from over 40,000 customer vehicles to lay the groundwork for realizing robust requirement engineering. A systematic method for analyzing big data on the cloud was introduced. The derived component-specific damage distribution from these collectives adopts a unique approach, utilizing the 1% vehicle term instead of the common 1% customer term to represent typical customer stress. This study shows that the driven mileage and the number of vehicles are crucial factors in 1% vehicle analysis. An analysis of the characteristics of the 1% vehicle is conducted, followed by an exploration to determine the required vehicle quantity for obtaining stable results. The shape parameter of the damage distribution determines the necessary number of vehicles for a reliable conclusion. Additionally, a comparative analysis of market-specific customer requirements between the US and Europe is presented, and real usage differences in customer operations are explained using an operating point frequency heatmap. The information presented in this paper provides valuable input for optimizing durability design and conducting efficient, customer-oriented tests, resulting in significant reductions in development time and costs.

**Keywords:** battery electric vehicle drivetrain; durability; damage distribution; 1% vehicle; market-specific requirements

## 1. Introduction

The battery electric vehicle (BEV) has become a central focus in automotive industry development. Electric vehicles are considered an excellent alternative to internal combustion engine vehicles (ICEVs) due to their unique properties. These include lower well-to-wheel (WTW) costs [1], zero local emissions, higher powertrain efficiency, and better driving performance. Automobile manufacturers strive to deliver top-tier products with innovative features to meet customer demands and enhance overall customer experience. At the same time, the industry aims to streamline development processes, reduce time and costs, and secure a more significant market share in an intensely competitive environment. It is crucial to design the drivetrain with a focus on efficiency and durability from a customer perspective to gain a comprehensive understanding of real-world vehicle usage. This approach ensures

customer satisfaction by preventing the pitfalls of undersizing or oversizing components. A design and testing strategy grounded in customer data is an efficient method for balancing customer expectations and design standards. Concurrently, the ongoing digitalization in the automotive industry has led to an increasing number of connected vehicles capable of cloud communication, which facilitates the wireless collection of customer driving data. Cloud data analytics is a versatile tool that offers multifaceted applications in achieving data-driven development objectives. Innovations such as digital twin technology and predictive maintenance, enabled by the intelligent monitoring of component conditions, become viable possibilities. Figure 1 presents a schematic overview of the main pillars guiding the utilization of powertrain data. The development trajectory of the drivetrain is envisioned to be holistic, systematic, and inherently customer-oriented, as introduced in [2]. This statement reflects the industry's dedication to meeting changing demands and adopting innovative technologies.

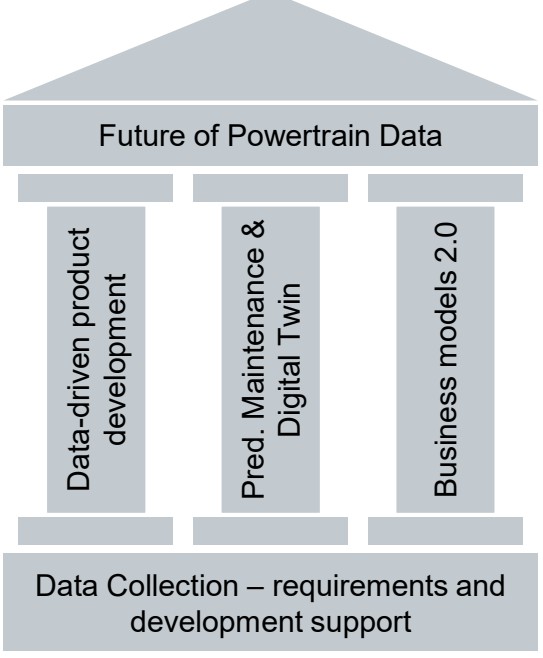

**Figure 1.** Future of powertrain data.

In recent years, the focus of further BEV drivetrain development has been on batteries, energy management strategies, control methods, and cost analysis. [3–6] The development of drivetrain durability presents new challenges due to the growing number and complexity of electrical components. The lifetime of the drivetrain, a complex system comprising multiple subsystems and components, is determined by the lifetime of each individual component. The components can be divided into two subsystems: firstly, electrical systems, including the battery, power electronics, and electric machine (EM); and secondly, mechanical systems, such as the transmission, differential, drive shaft, or cardan shaft. Electrical components are primarily affected by electrical stress, such as changes in voltage or current. Additionally, temperature (both ambient and operating) plays a crucial role in the aging of electrical components due to thermal stress. Related research has been carried out for traction battery durability [7,8], EM durability [9–11], power electronics [12], and the electric system [13]. The mechanical components in electric vehicles are subjected to more critical working conditions than those in conventional vehicles with multi-stage transmissions. This is due to EM properties such as the high-frequency current [14–16], higher speed range, and recuperation. Recuperation with high negative torque, for example, leads to additional stress on torque transmission components such as gears, shafts, and cardan shafts. The authors of [17] demonstrate an innovative approach by combining load spectrum calculation and condition monitoring to improve the precision of lifetime

prediction for gearboxes in BEV. A digital twin is considered a tool that enables the further optimization of the efficiency and reliability of BEVs and offers a high reliability design for powertrains. [18] Common norms, such as those for EM [19,20] and gear [21,22] development, are also used to achieve a robust design. Previous testing strategies must be adapted to address the new challenges of drivetrain testing and testing technology [23].

A crucial use of customer data is in durability design, which requires determining representative customer load collectives, or load spectrums, to ensure component strength for a defined desired lifetime. Representative customer load collectives are typically defined as load collectives that cause damage or stress to one or more customer vehicles, exceeding the load collective experienced by the remaining 99% of customers. The term '1% customer' is often used to refer to this definition, which can be obtained by analyzing the statistical distribution of customer loads, provided that sufficient data are available. Acknowledging the '1% customer' is a fundamental task in durability design. Load spectra can typically be derived directly from customer-related measurements, often collected using dataloggers [24–26]. However, conducting measurements with a large sample size can be costly. Additionally, the load spectrum of an individual customer vehicle or a small pool of test samples may not provide a representative reference load, particularly in the mass-produced passenger car field. This is due to the significant variation in customer driving behavior, which is the main cause of component damage, and the driving environment, which cannot be adequately described with a small amount of data. Onboard classification with aggregated data was frequently utilized, particularly in chassis and body durability analysis [27–32]. These data are classified and stored in vehicle control units and then retrieved through wireless communication or by the workshop. This method of data collection has great potential for acquiring data from a large number of customer vehicles. However, the depth of information is abstracted due to the classification, resulting in a loss of time series.

Several approaches exist for simulating representative customer load collectives, such as the 3D method (3D: driver, driven vehicle, and driving environment) developed by the Institute of Automotive Engineering at the Technical University of Braunschweig. This method, which was first introduced in [33,34], provides a detailed and practical description of customer usage. It was used to determine representative load collectives for gears and bearings in gearbox development. Its function and application were expanded to systematically, virtually, and holistically develop and design drivetrain and chassis components, considering representative customer use or collectives [35–40]. Representative load collectives are obtained via simulation, using a comprehensive database that identifies the typical driver, driving environment, and vehicle combinations (also known as parameter space). The first step in building the 3D database is to measure the target vehicle in the target market. At the same time, the necessary parameters for the target components are selected based on requirements. The measurement data are processed to characterize the 3D parameter space and generate statistics for the simulation. The load collectives of the combination with the most intensive loads are then simulated for the representative load collectives. This allows for obtaining component-specific representative load collectives. Refer to [41] for more information on this method. The Fraunhofer Institute for Industrial Mathematics has developed a method for simulating representative customer collectives using the Monte Carlo method [42–44]. By calculating the total damage to individual users, the most critical combination of load cases for a user population can be identified. The simulation requires systematic measurement as a foundation.

The introduction indicates that there has been limited research on customer-oriented durability development in BEV drivetrains, especially regarding mechanical components. Investigations into durability design based on representative load collectives are often restricted by sample size and rely on simulation. Therefore, it is necessary to use new technology to conduct big data durability analysis. The digitalization of the automotive industry has made it possible to collect online data from connected vehicles. This wireless collection of customer data with customer content enables the analysis of customer usage

of vehicles and their components. This paper investigates representative customer load collectives using cloud data obtained from over 30,000 customer vehicles. It introduces a systematic analysis method for cloud data and applies it to derive load collectives for each vehicle. Damage is calculated using these load collectives and component-specific Wöhler parameters, leading to the establishment of statistical damage distribution. A groundbreaking approach is introduced by defining the 1% vehicle as a benchmark for representative load collectives, diverging from the conventional 1% customer definition. The analysis reveals two important factors: driven mileage and vehicle quantity, shedding light on their impact on the 1% vehicle and its associated damage. This study also includes a comparative analysis at both the component and vehicle levels, highlighting typical properties of the 1% vehicle. This paper also includes an investigation to determine the necessary quantity of vehicles for a meaningful statement in durability analysis, which is dependent on the shape parameter of the distribution. Additionally, it presents a comparative examination of market-specific customer requirements between the US and Europe. This paper presents a novel method for conducting durability analysis in drivetrain development. The results obtained from representative samples can be used as design standards and to optimize durability design, as well as to conduct efficient and customer-oriented tests. Furthermore, the method and results provide a basis for implementing innovative functions, such as predictive maintenance.

## 2. Technologies for Data Collection in Customer Vehicles

Various technologies exist for collecting driving data. A review of traditional data collection methods was presented by [32]. Online data collection has become increasingly popular due to its advantages in sample size, configurability, processability, and efficiency [45]. Wireless data collection and transfer are also suitable for mass-produced vehicles. Figure 2 illustrates the general functional principle of online data collection. Typically, a central control unit communicates with a cloud server and multiple electronic control units (ECUs). The system can receive signals from a controller area network (CAN-Bus) or other bus systems and can send the signals to the cloud via telemetry. This configuration allows for defining the considered vehicles, signals, and frequency for raw data collection, providing flexibility for multi-purpose data collection.

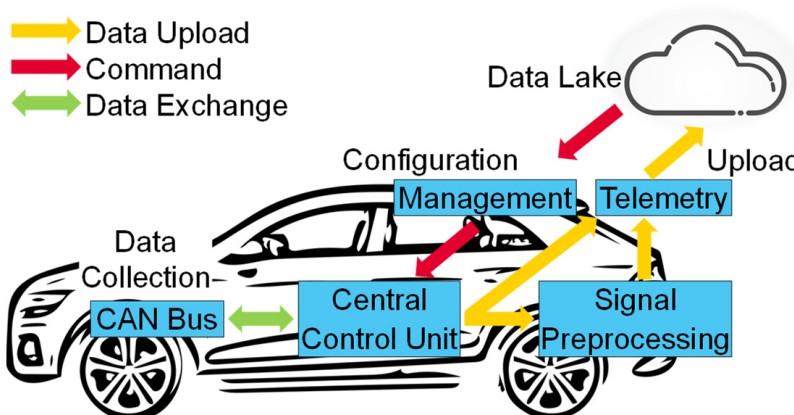

**Figure 2.** Online data collection in a vehicle.

Vehicles equipped with wireless communication functions (over-the-air function, OTA-Function) collect data in two main ways: raw data as time series and onboard aggregated data, as shown in Figure 3. The raw signals, or signals after simple preprocessing, can be uploaded to a data lake via telemetry as time series and evaluated using cloud analytics tools. The signals must always be collected with a timestamp for each sample. This data collection method provides a high level of information, which can be used, for example, to describe driving operations. However, it can result in large amounts of data and incur costs for transfer, storage, and computing resources. Therefore, due to technical and economic

limitations, the data frequency transferred is usually lower than the original frequency. Low-frequency data can still be utilized for various research purposes. The significance level of low-frequency data in the development of drivetrain durability was investigated in [45]. Data collected in this manner were used as a database for this paper. Raw data can also be aggregated on board using various preprocessing, identification, classification, or calculation methods, similar to the onboard collection function. The choice of classification method and signals depends on the use case. The results of these online classifications, which capture the most frequent data, will be sent to the cloud. The results are easily usable for further analysis in the backend. This online data collection method allows for the gathering of anonymous driving data with the customer's consent. These data can be linked to each vehicle, but the corresponding interpretation, including information about the driver, driving behavior, and driving environment, is only allowed within the scope of the General Data Protection Regulation (GDPR) law.

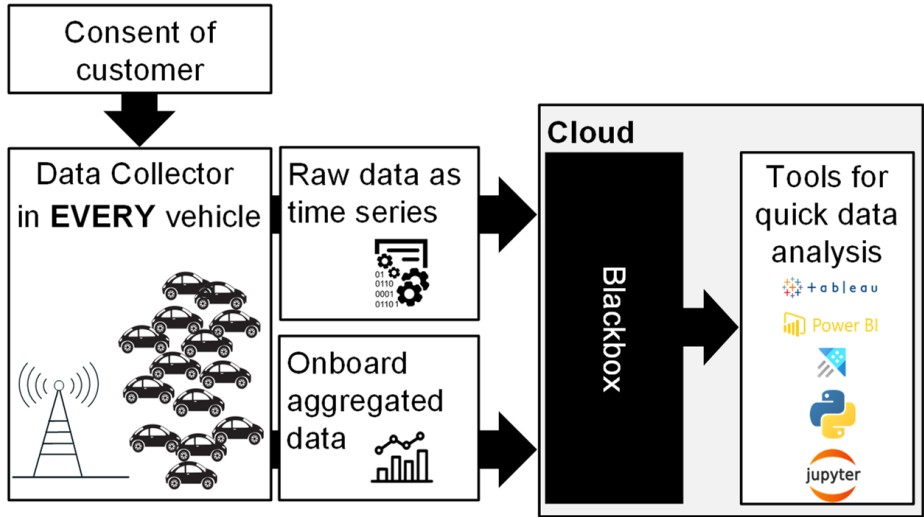

**Figure 3.** A visual of customer data analysis.

## 3. Drivetrain Durability Design

The primary objective of durability design is to ensure the operational reliability of the product, including vehicles, systems, and components, to prevent failure during their intended lifetime. Drivetrain durability development is based on research into structural durability, as introduced in [46–48]. In [49], the authors present special considerations for analyzing drivetrain durability, including fundamental notices and statements about the damage mechanisms of mechanical components, as well as component-specific calculations in design. This study also outlines the procedure for reliability analysis and measures to increase reliability, such as processing methods in production and typical durability testing possibilities. Durability refers to a component's ability to withstand various mechanical loads, such as cyclic or sudden loads. Additionally, the load on the electric drivetrain can be either electrical or thermal.

Figure 4 illustrates the relationship between customer stress and component strength in durability design. Both distribution frequency and stress are represented logarithmically, with frequency on the Y-axis and stress on the X-axis. The variation on the customer stress side is caused by diverse types of vehicle use. Customer stress distribution is dependent on driving behavior, the driven vehicle, and the driving environment [41]. Component strength variation is primarily due to inhomogeneities in material properties and manufacturing influences. To ensure a technically safe design, it is necessary to guarantee that the required component strength, with its known distribution, is sufficiently distant from or has little overlap with the stress experienced during customer operation. This often requires the component to endure stress corresponding to the 1% quantile in customer operation with a probability of 99.9%. In this context, the term '1% customer' is often used to refer to a

customer who experiences damage or stress higher than the remaining 99% of customers. In the German automobile industry, it is expected that the drivetrain should be able to withstand a mileage of at least 300,000 km without failure, even for the 1% of customers who exert the most stress on the product [50–52]. This expected target mileage nowadays depends on the intended application of the vehicle. For example, for a BEV as a small-city vehicle (often a second car), the target mileage can be set at 150,000 km [53]. However, for taxi use or ridepooling, the target mileage is often up to 600,000 km. Meeting requirements and market-specific design philosophy are both important factors. Therefore, a market-specific analysis of driving behavior and resulting customer collectives is a prerequisite. To design drivetrain components or systems without flaws and oversizing, it is necessary to determine the representative 1% customer based on knowledge of customer demands derived from customer usage data.

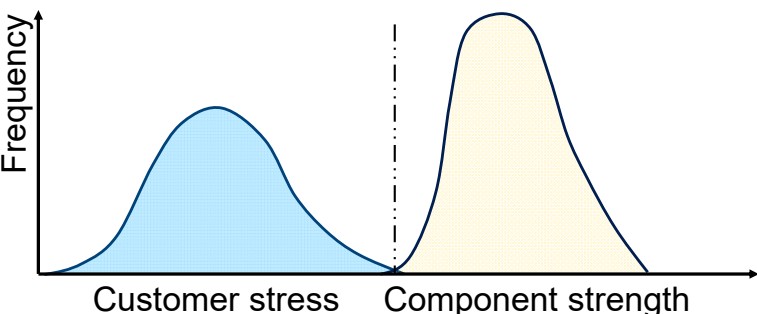

**Figure 4.** Customer stress and component strength.

Figure 5 illustrates the process for optimizing a drivetrain durability design based on customer data. The first step is to evaluate customer data to obtain real-life vehicle usage in the form of load collectives. Simultaneously, the actual requirements in development must be analyzed, including design and testing standards, based on the experience of previous vehicles. This approach should approximate real customer usage. After comparing customer requirements with design standards, it is then possible to identify optimization potential and define new target requirements. These target requirements can be used as design and test specifications. Furthermore, it is possible to make an objective comparison between different markets and products.

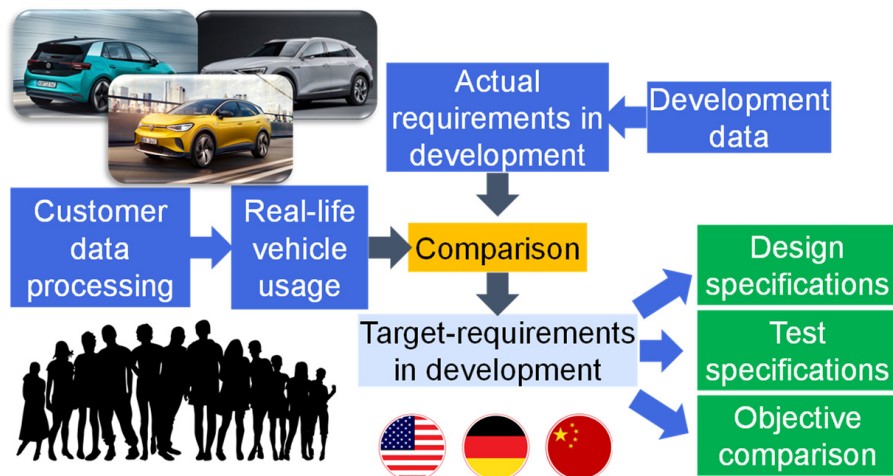

**Figure 5.** Optimization based on customer requirements.

In the entire process, it is important to first create a clear overview of the damage mechanisms of the respective components. The service lifetime of the drivetrain, a complex system consisting of numerous components and subsystems, is determined by the lifetime of each individual component. Several commonly used classification methods exist for

obtaining mechanical, electrical, and thermal load collectives from driving data. This investigation focuses solely on rollover classification, which is illustrated in Figure 6. Rollover classification is typically used for rotating components such as gears, bearings, and shafts. Rollover classification necessitates a load signal, such as torque, in specific classes, as well as a rotational speed signal. It explains how many rotations a component has undergone due to the torque within a defined time interval, $\Delta t$. To determine the number of rotations based on the measured rotation speed, it should be divided by the sampling time. The rollover number for a certain torque class, *i*, can be calculated using the following equation for all speed samples within this load class:

$$n_i = \sum \frac{Speed_i}{60} \cdot \Delta t \tag{1}$$

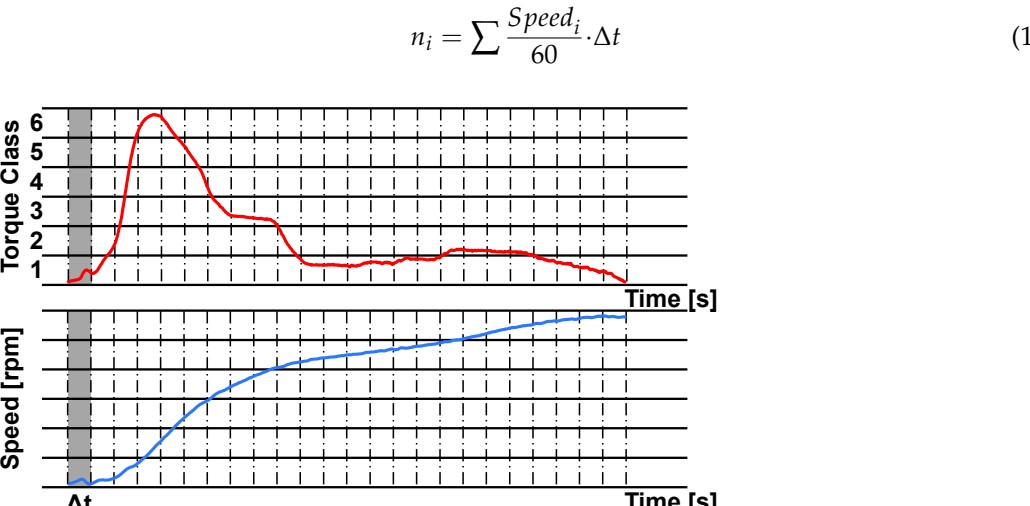

**Figure 6.** Rollover classification with torque class (red) and speed (blue).

To estimate the damage and corresponding lifetime, linear damage accumulation is performed using the Wöhler line [54] and load collectives [55,56]. Figure 7 displays an exemplary double logarithmic Wöhler line and a load collective. $L_D$ and $N_D$ represent the load class and the number of load or load cycles of fatigue strength range, respectively. $L_i$ denotes the load of one load class, while $N_i$ and $n_i$ are the corresponding load counts that occur on the Wöhler line and in operation. Component-specific considerations are necessary when evaluating collectives with Wöhler parameters, such as the inclination of the Wöhler line or fatigue strength.

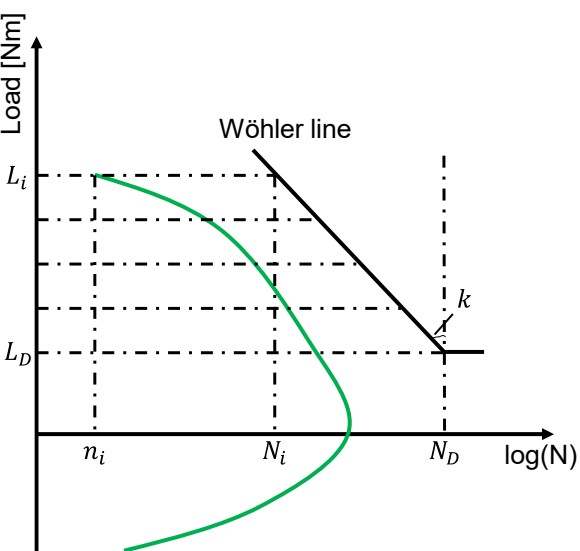

**Figure 7.** Wöhler line for lifetime calculation with torque as load.

The exponent *k* determines the inclination of the Wöhler line, which can be described by the following equation:

$$N_i = N_D * \left( \frac{L_i}{L_D} \right)^{-k} \tag{2}$$

The cumulative damage D can be calculated with the following equation according to linear damage accumulation [53,54]:

$$D = \sum_i \frac{n_i}{N_i} = \sum_i \frac{n_i}{N_D} * \left( \frac{L_D}{L_i} \right)^{-k} \tag{3}$$

## 4. Investigation of 1% Vehicle

Small amounts of data are insufficient to adequately describe representative customer collectives. To systematically collect and analyze data, it is necessary to first define the required signals. To interpret customer driving behavior, it is important to consider the accelerator pedal position, brake pedal position, recuperation level, selected driving mode, and vehicle speed, with simultaneous acceleration and data fusion. These can provide descriptive driver statistics, although individual driver identification is not possible. To reconstruct driving conditions, various signals such as environmental temperature and slope are required. Signals such as rotation speed and torque (EM/Wheel), temperature (EM/Power electronics/Battery/Oil), voltage, and current (EM/Power electronics/Battery) from the components can be directly processed to determine damage or stress.

Each active vehicle can save collected data for a defined duration or mileage as a data package. This interval is referred to as a trip, which is the basic unit for further analysis. Figure 8 illustrates that each data package contains time series and statistical information, including frequency distributions, operation points, and load collective. By creating a data matrix for each trip and each available vehicle in the market, a complete description of component usage can be established. Combining all trips with a long enough duration of one vehicle into a single package allows for a comprehensive analysis of component usage. With enough data from multiple vehicles, it is possible to derive the distribution of damage and identify critical situations, such as the top 1% load collectives. Typically, a vehicle is used by multiple drivers. It is not feasible to directly identify individual drivers through existing cloud data. Therefore, the vehicle is a more appropriate unit for describing vehicle usage and its collectives, rather than the driver. As a result, the '1% customer' term can be revised to the more reasonable '1% vehicle'. This revised term maintains the same technical meaning and content; 1% vehicles are those with damage higher than 99% of the remaining vehicles. Similar definitions, such as 5% and 10% vehicles, can also be made. The damage mechanisms of the components are different, leading to component-specific 1% vehicles. Additionally, the 1% vehicle is market-specific and model-specific due to differences in usage.

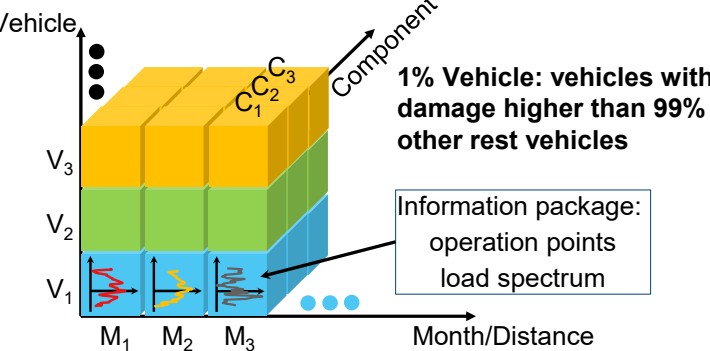

**Figure 8.** Data analysis for 1% customer collectives.

For this investigation, an electric SUV was chosen. This SUV has two versions in Europe: a basic version with less power and battery capacity, which serves as the reference vehicle, and a performance version with more power and battery capacity, which is the main focus of this paper. In the US, only the performance version of the car is available. The drivetrain and its components are identical between the EU and US versions. All vehicles are equipped with an all-wheel drive, consisting of a primary EM on the rear axle and a secondary EM on the front axle. Each shaft is equipped with a single-speed gearbox specifically designed for the EM.

A database of 10,812 performance vehicles and 14,066 basic vehicles in Europe, as well as 15,508 vehicles in the US, was used to evaluate driving data in the backend. Figure 9 displays the probability distribution of annual mileage for these vehicles. To ensure a fair comparison, the driven mileage of each vehicle was scaled to the same period due to varying measurement durations. It is worth noting that vehicles with an annual mileage exceeding 60,000 km are uncommon. To ensure accuracy, the figure limits annual mileage to 60,000 km. The majority of vehicles in Europe (blue for performance and green for basic) have an annual mileage between 6000 km and 18,000 km. The average mileage is approximately 11,000 km, with a median mileage of about 10,000 km. According to the German Federal Motor Transport Authority, the average annual mileage per car in 2020 was approximately 13,323 km [57]. By contrast, US customers have a higher annual mileage. The average mileage is approximately 14,000 km. The median mileage is similar to Europe at 12,000 km. According to [58], the average mileage for electric cars in the US is approximately 11,400 km, while electric SUVs have an average mileage of around 16,400 km. Annual mileage depends on various factors, such as usage area, scenarios, vehicle type, and the drivetrain. The basic version of performance vehicles in Europe generally has a lower annual mileage due to a smaller battery and shorter operating range.

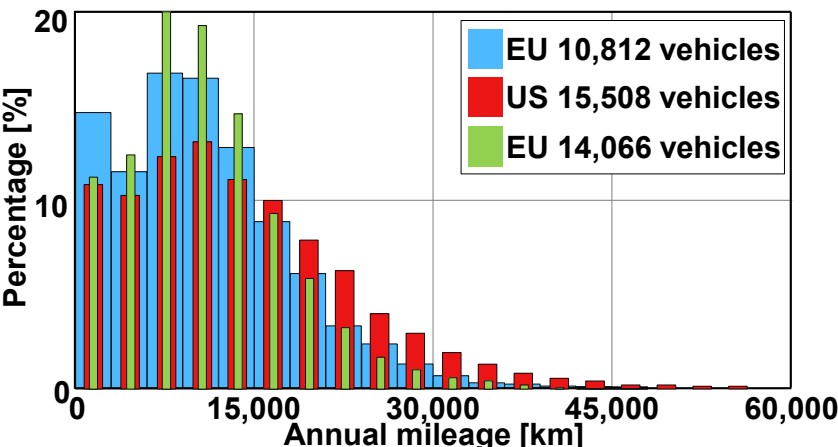

**Figure 9.** Annual mileage distribution in Europe and the US.

To conduct an analysis of the 1% vehicle, the data for each vehicle were classified to create load collectives. These collectives were then scaled to a standard mileage of 300,000 km for comparison. Using these collectives, the damage to each component was calculated based on component-specific Wöhler parameters. The components selected for analysis were the gear with tooth root and tooth flank, bearing, and shaft on both the rear and front axles. The classification of the shaft was based on the wheel torque and wheel speed, while the torque and speed of the EM were used for the other components. After damage calculation, statistical damage distribution could be determined to identify the 1% vehicle. Annual mileage was used as a selection criterion to choose effective vehicles. The minimum annual mileage threshold affects the number of effective vehicles, which is a key factor in determining the 1% vehicle. Finally, only the effective vehicles should be analyzed to determine the 1% vehicle.

Figure 10 shows the damage distributions after filtering with 12,000 km for the gear tooth flank on the front axle. The right part of the figure shows a typical damage distribution graph with the damage as the logarithmic X-axis and the frequency as the Y-axis, similar to a normal distribution. Distribution fitter functions can be used to obtain distribution parameters such as scale and shape parameters of the Weibull distribution. Percentiles can be calculated to determine the damage of the 1% vehicle based on this distribution and its positioning.

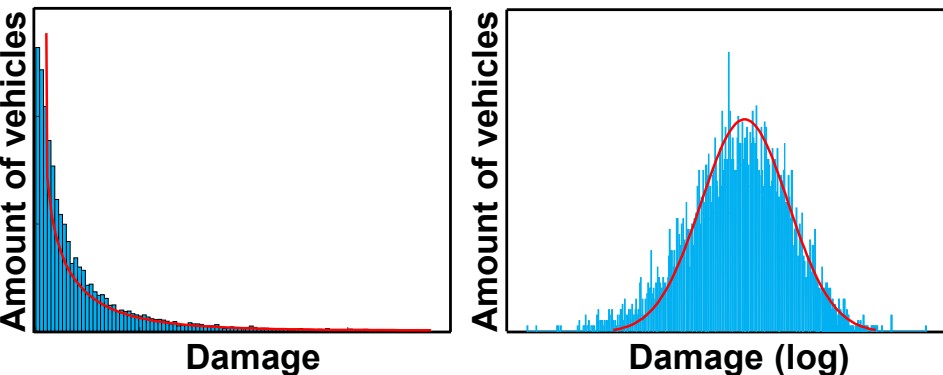

**Figure 10.** Distribution of damage among vehicles fitted with Weibull or Gaussian distribution (red lines).

Figure 11a illustrates the impact of driven mileage on the 1% vehicle and its damage to the tooth flank on the front axle. The damage of the 1% vehicle with almost all the vehicles (selection criteria: mileage greater than 0 km) is significantly higher than that of the other annual mileages due to the possible special driving situations, such as test drives, which often involve high loads to detect vehicle performance limits. As previously mentioned in [45], drivers typically require an adaptation phase to become accustomed to driving a new vehicle. During this period, the collected data on damages may not accurately reflect normal driving behavior. Additionally, seasonal influences may also impact the data collected. Therefore, the information on damage from these vehicles is not statistically stable and cannot be used for further analysis. This impact can be mitigated by increasing the distribution density, which compensates for the increased number of vehicles analyzed.

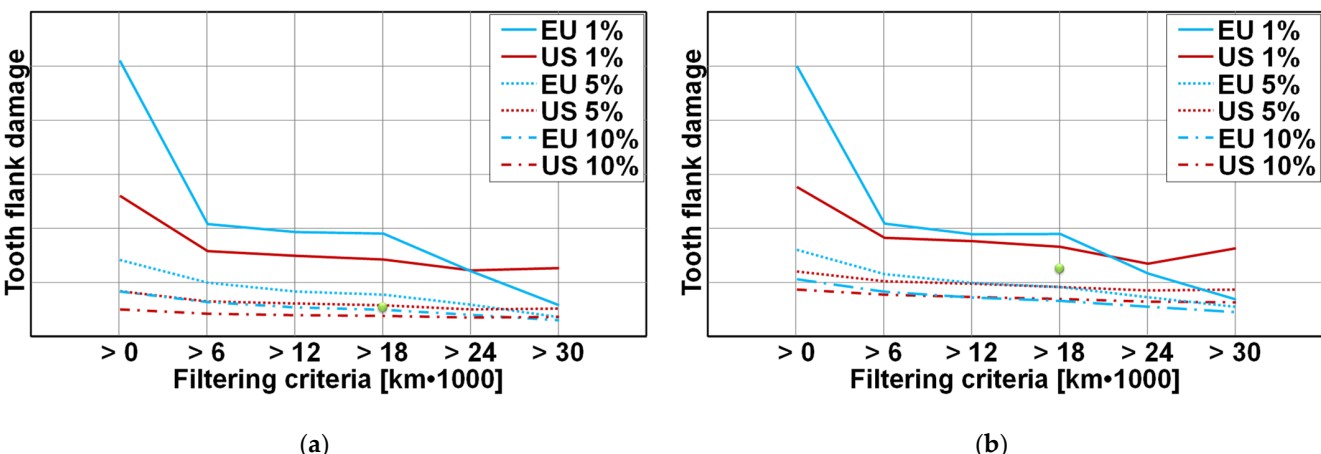

**Figure 11.** Impact of mileage on the 1% vehicle of the tooth flank. (**a**) Impact of mileage on the 1% vehicle of tooth flank on front axle; (**b**) Impact of mileage on the 1% vehicle of tooth flank on the rear axle.

When using annual mileage of 6000 km, 12,000 km, and 18,000 km as selection criteria, the damages of the 1% vehicle remain at a similar level. However, if the vehicle amount

is reduced due to a higher mileage demand, the damage of the 1% vehicle decreases, particularly in Europe. This is because the number of selected vehicles falls below a certain threshold. After selecting European vehicles with an annual mileage of at least 24,000 km, there are fewer than 600 vehicles remaining, which can lead to unreasonable statements.

Another reason for the lower tooth flank damage in vehicles with higher annual mileage (over 40,000 km per year) could be attributed to the frequency of highway driving, which typically involves lower torque demand. This relationship is illustrated in Figure 12. This figure indicates all the vehicles in the EU and the US with a driven mileage of over 6000 km and up to 60,000 km. This figure shows that the high damage vehicles are the ones with lower annual mileage. This can be due to driving maneuvers, such as more city driving with acceleration and deceleration. It is more likely that vehicles with a low annual mileage are driven in an urban scenario with many start/stop maneuvers. When the vehicle shows a higher annual mileage, such as for vehicles with more use on the highway, the damage remains low because of driving maneuvers with lower torque. During a start/stop maneuver in city driving, the torque is much higher compared to when the power reaches the power hyperbola.

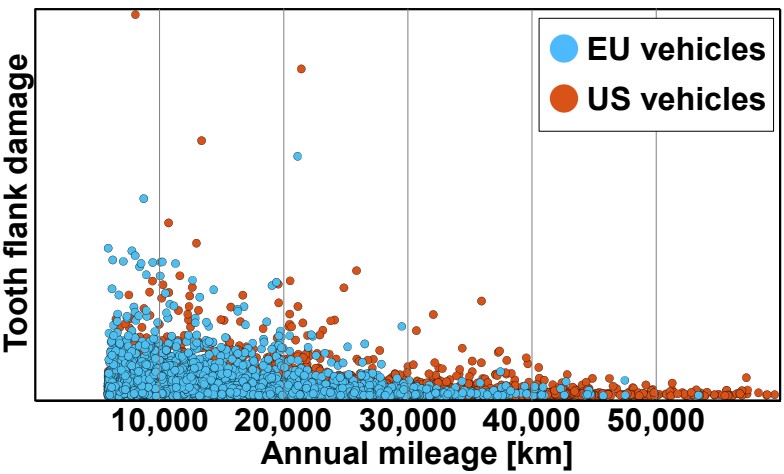

**Figure 12.** Relationship between tooth flank damage and annual mileage with an annual mileage of more than 6000 km.

When comparing the damage of the 1% vehicle in the EU to that of the 1% vehicle in the US, it is evident that the damage in the US is significantly lower, at approximately 74.9% of the damage in Europe. This suggests that the customer requirements for the tooth flank of the front axle in Europe are higher than those in the US. When analyzing vehicles in Europe and the US, combining them into one database can reduce the damage caused by the 1% vehicle due to the lowering effect of the US vehicles. However, in Figure 11a, where there are more vehicles at the lines of the 1% and 5% vehicles in the US, the decrease is not as clear. The damage caused by the 1% or 5% vehicle remains almost constant. Using 18,000 km filtering as an example, the damage caused by 5% of vehicles in Europe is approximately 41.1% of that caused by 1% of vehicles. The same ratio applies to the US, at about 41.2%. The damage caused by the basic version's 1% vehicles, represented by the green point in this figure, is almost at the same level as the damage caused by 10% performance vehicles in Europe. This provides evidence for the design philosophy of differential treatment. Thus, the basic version was not specially analyzed in this paper further.

Figure 11b displays the 1% vehicle damages of the tooth flank on the rear axle in relation to the driven mileage. The line trend is similar to that of Figure 11a. The vehicle is driven by two axles due to the all-wheel drive being applied by the defined torque distribution map. The primary torque source is responsible for the majority of the wheel torque. It is worth noting that the 1% vehicle damage in the US is 87.4% compared to Europe, which is attributed to different driving behaviors. The 5% and 10% vehicles are in

the same position at 12,000 km, indicating similar customer requirements. The tooth root results are comparable due to the similar Wöhler inclination. Therefore, the results are not presented separately in this paper.

Figure 13a displays the 1% vehicle damage of the bearing on the front axle in relation to the driven mileage. Despite different Wöhler parameters, the 1% vehicle damage in the US is 95.4%, almost identical to that in Europe. Figure 13b illustrates the 1% vehicle damage of the bearing on the rear axle. The 1% vehicle damage in the US is 115.3%, higher than the value in Europe. The stability of the damage line course in Europe is primarily influenced by the driven mileage.

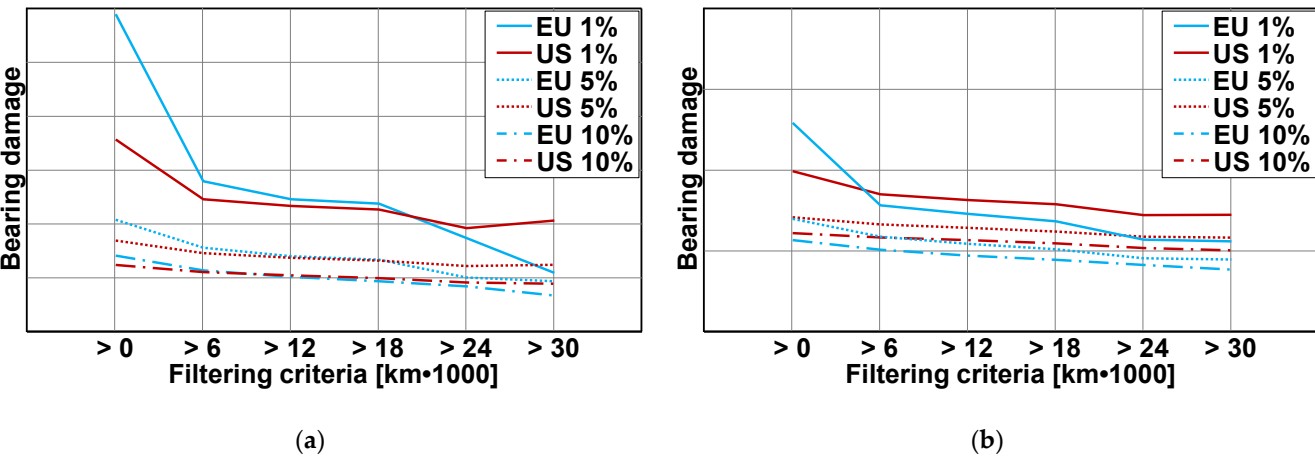

(**a**)                                                                     (**b**)

**Figure 13.** Impact of mileage on the 1% vehicle of the bearing. (**a**) Impact of mileage on the 1% vehicle of the bearing on the front axle; (**b**) Impact of mileage on the 1% vehicle of the bearing on the rear axle.

Figure 14 presents the damages of the front and rear shafts, which are derived from the wheel speed and shaft torque. The difference between the EU and US is not significant, especially for the rear axle after 12,000 km. It is important to note that the relationship between shaft damage and all-wheel drive strategy is dependent.

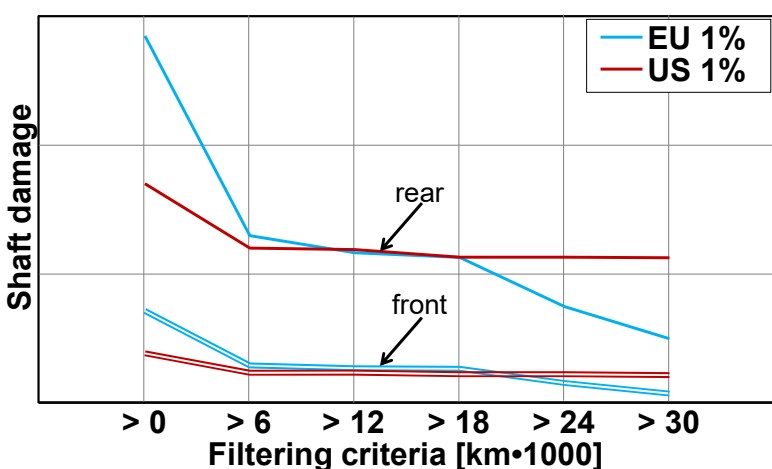

**Figure 14.** Impact of mileage on the 1% vehicle of shafts.

Typically, one vehicle at the 1% vehicle position with its collectives can be found. However, to reduce randomness in customer usage, it is recommended to select several vehicles closest to the 1% vehicle (99th percentile) to build the overall collective. For instance, the target can be defined as four vehicles from both sides of the 1% vehicle position, including the exact 1% vehicle. Alternatively, more vehicles can be considered, or only vehicles from the right side of the 1% vehicle position. However, if the vehicles

only come from one side, it may lead to undersizing or oversizing. Figure 15 illustrates an example of 1% vehicle collective of the tooth flank. The five collectives differ due to their driving behaviors. The red vehicle is driven with more maximum torque and less medium load range compared to the other vehicles. The damage is almost the same for all the collective forms. The black line represents the collective with average values for each torque class, defined as the 1% vehicle collective.

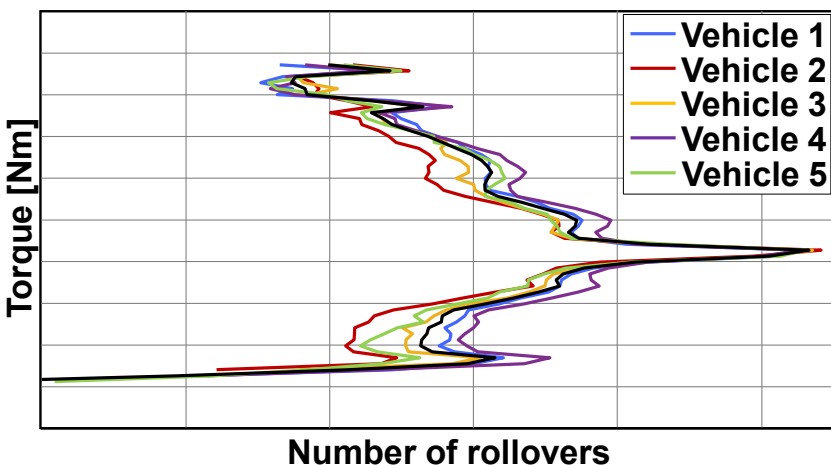

**Figure 15.** An example of 1% vehicle collective from five vehicles.

The results in this section indicate that the mileage driven and the number of vehicles are two factors that determine the 1% vehicle. It is important to note that the 1% vehicle is specific to each component and requires separate consideration.

## 5. Comparison between Component Level and Vehicle Level

There is often a debate about whether to target the collective at the component or vehicle level for drivetrain or vehicle testing in the development of durability. Previous studies [32,59] have made initial attempts at total vehicle and chassis development in this direction. To increase objectivity, other signals or parameters such as vehicle speed, acceleration, or acceleration/weight ratio, which can be transformed into component collectives, can be used for collectives at the vehicle level. The first approach at the component level results in precise component dimensioning and testing. In order to combine collectives for different components, a method is introduced in [35] that uses collectives as a subset when they do not have a cross point in their collectives, and overlapping collectives when they do have a cross point. However, this method requires comprehensive knowledge of each component, which may require more effort. The second approach at the vehicle level focuses on the vehicle or system. The implementation, especially in optimizing the testing cycle, is more straightforward. However, there is a risk of undersizing several components.

In terms of durability design for all-wheel vehicles, it is important to discuss the design of components on both the front and rear axles, taking into account specific customer requirements or worst-case scenarios at the vehicle level. Therefore, it is necessary to identify any discrepancies in the design from various perspectives. To clarify, this section investigates the 1% vehicle and analyzes its total wheel torque and wheel speed at the vehicle level. The fictitious Wöhler inclination 5, typically used in analyzing components with unknown Wöhler parameters, is used to calculate the damage at the vehicle level. The fatigue strength is based on a number of loads of $10^7$. This approach generates lines of fictitious damage at the vehicle level, as shown in Figure 16. The impact of driven mileage and vehicle quantity can be directly observed.

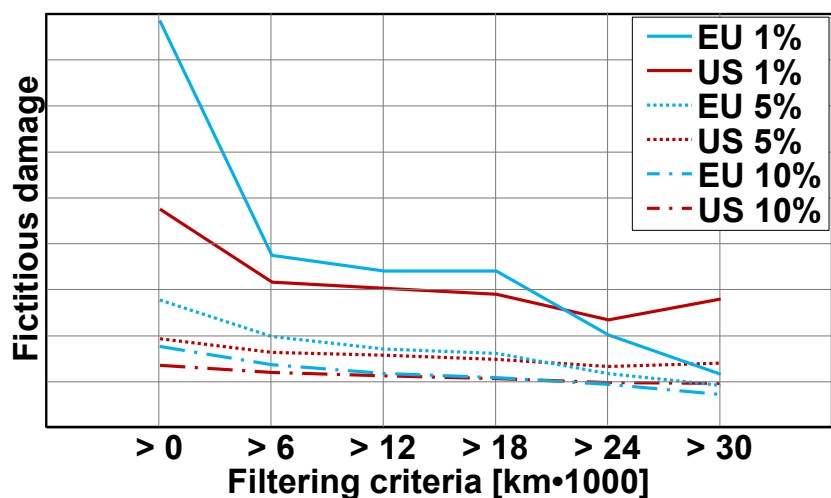

**Figure 16.** Impact of mileage on the 1% vehicle at the vehicle level.

The vehicles in Europe are used as an example of this explanation. In total, there are approximately 4000 effective vehicles that meet the 12,000 km filtering criteria. The basic requirement in design is that undersizing should be obviated. The vehicle closest to the 1% position at vehicle level, along with several other nearby vehicles, may not be stationary for each component of the 1% vehicle without undersizing. To clarify, we analyzed vehicles to the right of the 1% position at the vehicle level, whose mean is already less than the 1% vehicle. This can reduce the likelihood of damage to their components being lower than the damage to the 1% vehicle, thus potentially preventing undersizing.

The ratios of damage between the components of vehicles at the vehicle level and the 1% component-specific damage are calculated, as shown in Figure 17. Values over 1 indicate damages that are higher than the 1% value. In Figure 17a, only 0.5% of the total vehicles on the right side of the 1% vehicle position were selected to build the overall collective, ensuring that the design of all components meets the basic goal without undersizing. However, this results in an approximately 10% oversizing of the bearing on the front axle and the gear and bearing on the rear axle. The number of vehicles is a key factor that affects the statement in this section. By applying a filtering criterion of 18,000 km, only about 1600 vehicles remain. The same analysis was conducted on these vehicles, and the results are presented in Figure 17b. In this case, only 0.4% of the total vehicles on the right side of the 1% vehicle position were selected to build the overall collective. Therefore, the design of all components can meet the basic goal without undersizing. When the total number of vehicles is lower, the damage curves become more sensitive to changes in position and vehicle count. If slight oversizing is accepted in design, such as the pink line in Figure 17, the 1% vehicle at the vehicle level can also be used for component dimensioning. This can significantly reduce development time and costs. However, this statement depends on various factors, such as the number of vehicles or market-specific damage distribution. Additionally, the statement is only valid for all-wheel drive vehicles with permanent torque distribution between the front and rear axles.

To explain the results in Figure 17, the peculiarity of damage distribution through a sensitivity analysis was investigated, using the tooth flank of the front axle as an example. Each vehicle in the cloud was originally numbered in order. To create Figure 18, the damages of effective vehicles, after applying a filtering criterion of 12,000 km, are ordered numerically. Most vehicles only have low damage values until approximately the 4000th vehicle. The position of the 1% vehicle is indicated by the red line in the breakover region. This figure displays several data points adjacent to the 1% distribution. The number of vehicles on the right side of the 1% position depends on the total number of vehicles, which corresponds to the lines in the figures in part 4. The curve after the 1% line rises significantly, indicating the line's sensitivity. In Figure 18a, a change of only 0.1% from 99.1%

or 88.9% results in a 3.2% increase or decrease in damage. The position is altered across four vehicles, but the change rate remains stable. A 0.5% change results in a significantly higher increase of 18.94% compared to a decrease of 11.65%, as indicated by the damage curve line progression. The distribution density decreases as the number of vehicles decreases, as shown in Figure 18b. A 0.1% change results in a shift of only two vehicles. There are few vehicles to the right of the 1% vehicle position. A 0.5% change leads to a 25.56% increase.

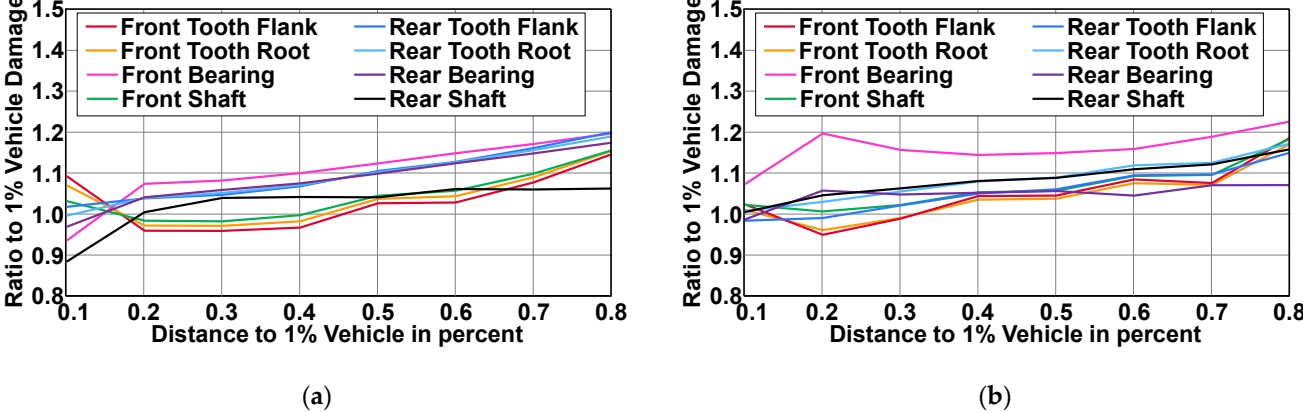

(**a**)　　　　　　　　　　　　　　　　　(**b**)

**Figure 17.** Ratio to the 1% vehicle damage with different vehicle amounts. (**a**) Ratio to the 1% vehicle damage with different vehicle amounts from right side of the 1% position—EU, 12,000 km as filtering; (**b**) ratio to the 1% vehicle damage with different vehicle amounts from the right side of the 1% position—EU, 18,000 km as filtering.

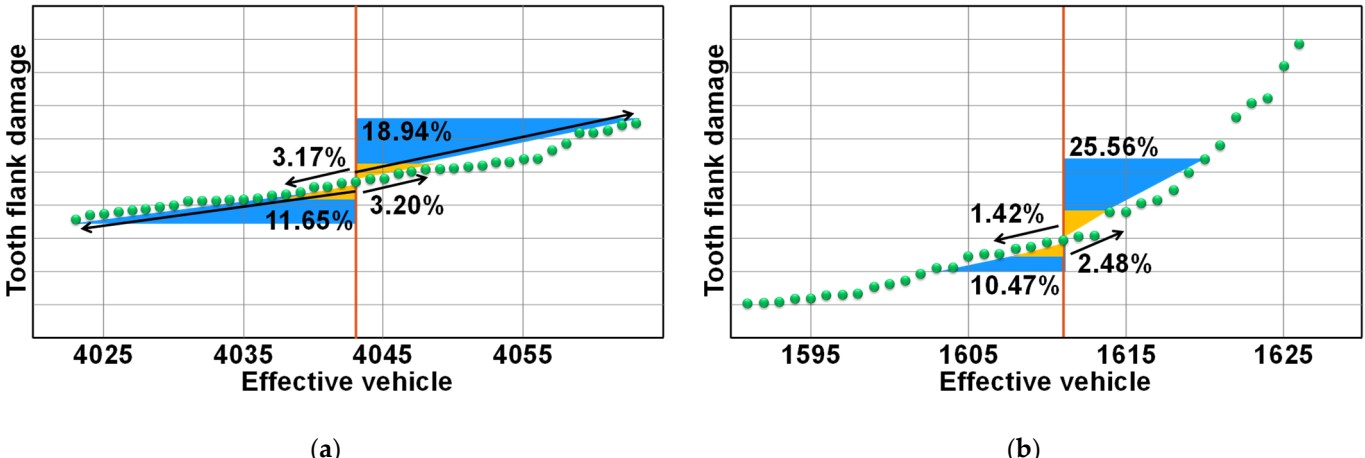

(**a**)　　　　　　　　　　　　　　　　　(**b**)

**Figure 18.** Damages arranged in order of numerical size. (**a**) Sensitivity of the damage curve with 12,000 km filtering; (**b**) sensitivity of the damage curve with 18,000 km filtering.

Table 1 provides an overview of the ratios of different percentages in comparison to the 1% vehicle damage for each component and at the vehicle level. The percentages are determined based on the distribution specific to each component.

**Table 1.** Damage ratio compared to the 1% vehicle damage of each component and at the vehicle level.

| Percent | 2% | 5% | 10% | 50% |
|---|---|---|---|---|
| Front-Tooth Flank | 72.4% | 43.5% | 28.4% | 5.4% |
| Front-Tooth Root | 72.2% | 45.3% | 29.7% | 5.9% |
| Front-Bearing | 80.8% | 57.3% | 41.8% | 12.4% |
| Front-Shaft | 72.9% | 47.0% | 31.3% | 6.7% |
| Rear-Tooth Flank | 79.0% | 52.7% | 38.5% | 13.5% |
| Rear-Tooth Root | 75.7% | 54.7% | 40.8% | 15.5% |
| Rear-Bearing | 88.0% | 74.8% | 65.1% | 39.8% |
| Rear-Shaft | 77.7% | 58.0% | 44.9% | 19.1% |
| Vehicle level | 70.9% | 47.7% | 32.2% | 8.9% |

## 6. Investigation of Needed Vehicle Quantity

As previously mentioned, the number of vehicles is a crucial factor in the analysis. To determine the sensitivity of this statement to the number of vehicles, the same investigation was conducted with a larger sample size. Specifically, the number of vehicles in Europe was increased to over 26,000, and in the USA, it was increased to about 22,000. Figure 19 illustrates that with a larger number of vehicles, the damage estimated for the 1% vehicle was similar to that of the 6000 km, 12,000 km, and 18,000 km scenarios because the initial vehicle number provided a significant level of statement validity. As the number of vehicles increased, the results for Europe when filtering with 24,000 km or 30,000 km were more reasonable. In the United States, the line position remained nearly unchanged. For annual mileages exceeding 18,000 km, the damage of the 1% vehicle was even more reasonable due to the increase in vehicle numbers.

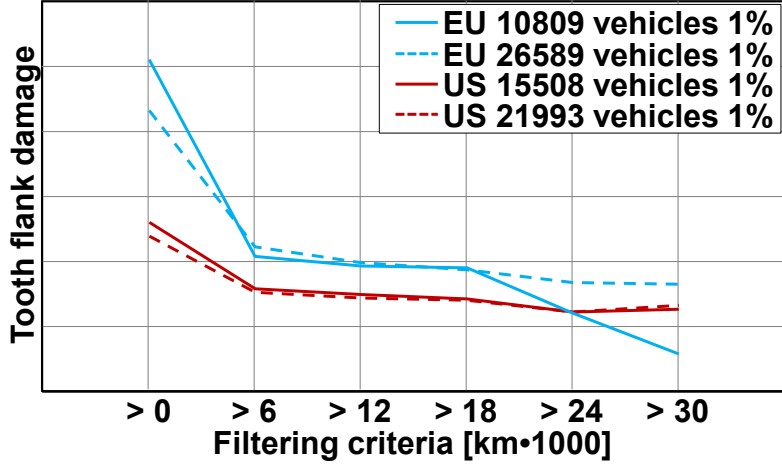

**Figure 19.** Impact of mileage on the 1% vehicle of the tooth flank on the front axle with additional vehicles.

In the study of the 1% vehicle, an important question arises: how many vehicles are required to provide a stable statement about the damage caused by the 1% vehicle? To answer this question, another investigation was conducted, as shown in Figure 20. The figure was drawn by selecting 100 to 10,000 random effective vehicles from those available in Europe. The damage caused by the 1% vehicle was calculated and illustrated on a chart for each iteration with varying amounts of random vehicles. The 10,000 iterations conducted in this investigation demonstrate the variance of the damage caused by the 1% vehicle with different vehicle amounts. The damage's 5–95% confidence interval (shown as boxes) should fall within the 90–110% range of the target damage, which is defined as the damage caused by the 1% vehicle, considering all available vehicles. In Europe, this translates to 4000 effective vehicles covering more than 12,000 km. According to the given probability distribution in Figure 8, a minimum of 10,000 vehicles should be defined for investigating the 1% vehicle. The required number of vehicles depends on the distribution's

standard deviation, shape parameter, and possible spread. It is important to maintain a balanced and objective approach, avoiding biased language and subjective evaluations. Despite the number of vehicles available in the market, approximately 4000 vehicles are still required due to the unique damage distribution with its shape parameter.

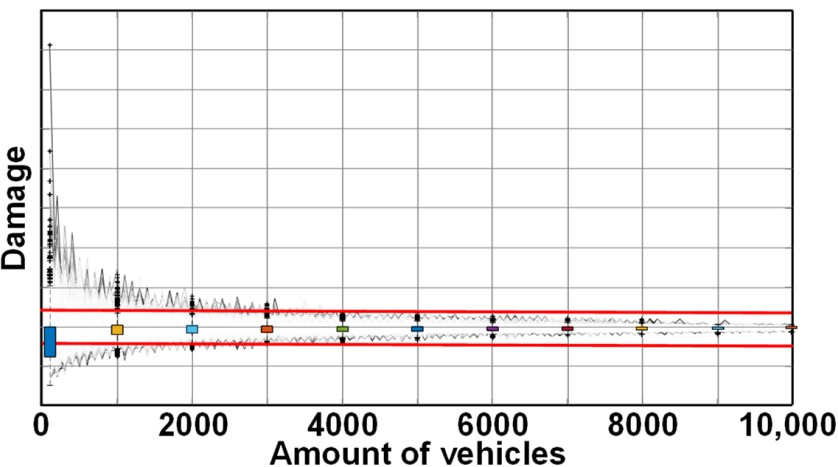

**Figure 20.** Needed vehicle quantity for a stable statement.

## 7. Market-Specific Requirements

As discussed in part 4 of this paper, the requirements for several components in the US are lower than those in Europe. Figure 21 compares the requirements for bearing, gear tooth flank, gear tooth root, and shaft between the 1% vehicle in the USA and the EU.

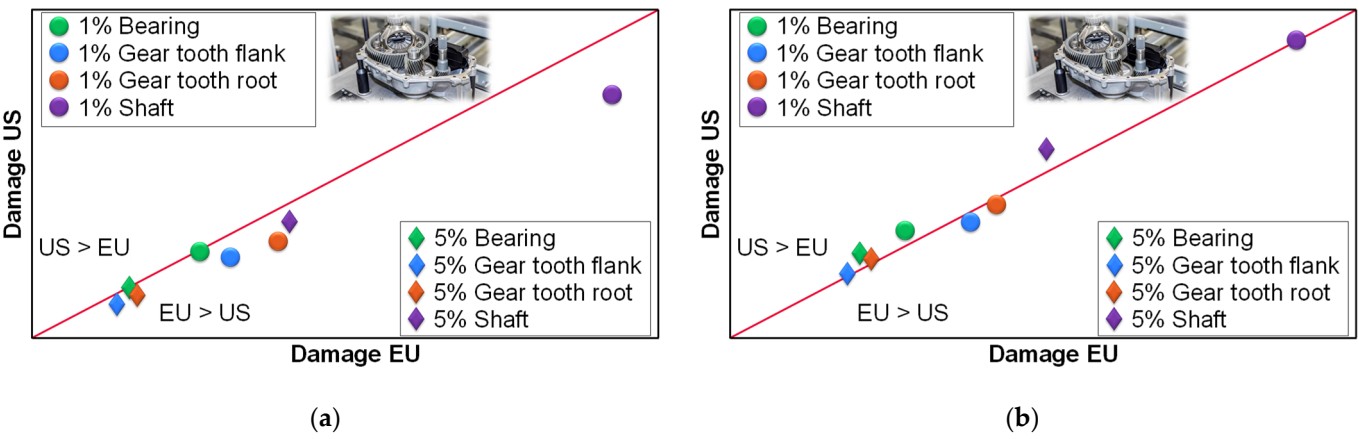

(**a**)                                                                 (**b**)

**Figure 21.** Market-specific customer requirements. (**a**) Market-specific customer requirements for components of the front axle; (**b**) Market-specific customer requirements for components of the rear axle.

In this figure, the damage caused by the 1% vehicle in the EU is higher than in the US below the red line. Due to different component-specific Wöhler parameters, for the bearing on the front axle, the damage of the 1% vehicle is precisely on the line, which is equivalent to that of the US, as shown in Figure 21a. On the other hand, for the bearing on the rear axle, the situation is reversed. Generally, the 5% vehicle requirements show a smaller difference between Europe and the USA.

The behavior of customers while driving is the primary cause of varying loads on vehicles and their components. Figure 22 displays a differential chart of the operating point frequency of customers in the EU and the USA. Customers in the EU dominate in the high acceleration and high-speed areas. The high acceleration area also typically results in high loads on the drivetrain. Due to torque distribution, when high torque is required, more

torque is delivered from the front axle, resulting in a higher load on the front axle in the EU compared to the US. This causes a higher rate of vehicle damage for EU customers. However, in the middle acceleration area, which is significant for bearing damage, US customers drive more frequently. Therefore, the rate of bearing damage for 1% of vehicles is nearly the same for both the EU and the USA.

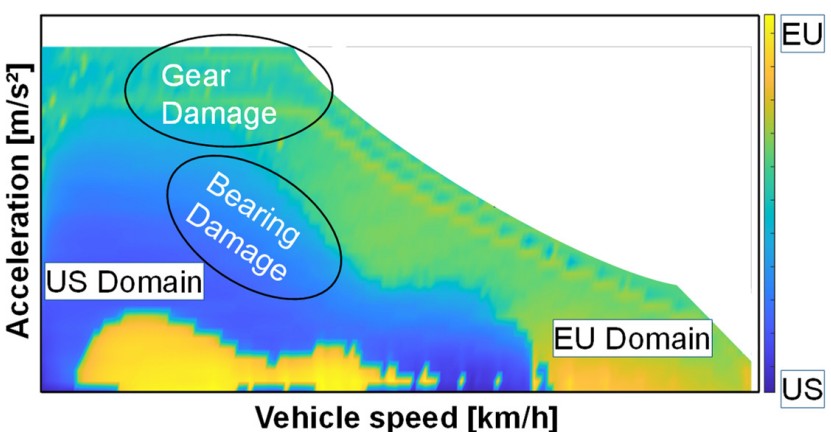

**Figure 22.** Operating point difference heatmap.

## 8. Conclusions

The derivation of load collectives being representative for customer data regarding durability design is a core issue in drivetrain development. This is the basis for the precise dimensioning and effective testing of drivetrain components. This study introduces a new method to obtain customer driving data by using online data collection. Driving data are collected from the vehicle and then sent to the cloud. This advanced technology not only facilitates robust data collection but also enables the analysis of extensive customer datasets in the cloud, establishing a fundamental framework for customer-centric development.

The procedure for systematically analyzing component design, with the trip as the basic unit, was defined based on the presented data matrix, taking into account the amount of data in the backend. By analyzing time series and statistical data, the lifetime usage of each component and vehicle can be determined for a sufficient amount of data. By comparing all vehicles, the distribution of damage can be determined to identify the top 1% load collectives. For the specific context of drivetrain durability design, the '1% vehicle' instead of the commonly used '1% customer', as defined, serves as a representative embodiment of customer requirements.

This paper investigates the performance and basic versions of an electric SUV in the EU and US. The database includes a total of 24,878 vehicles in the EU and 15,508 vehicles in the US, with only the performance version available in the US. We conducted a statistical plausibility check of the annual mileage data and derived collectives of the vehicles, which were then scaled to a standard mileage of 300,000 km for comparison. Statistical distribution was used to calculate the damage to the tooth root and tooth flank, bearing, and shaft on both the rear and front axles caused by those collectives. The derivation of 1% vehicles that are specific to components, models, and markets is a crucial aspect of our methodology.

The filtering criteria for selecting effective vehicles was based on driven mileage in order to avoid special driving situations that could excessively influence the 1% vehicle without being representative in the long term. The key parameters for understanding the 1% vehicle in this analysis are the number of vehicles and the driven mileage, which were determined by analyzing the damage lines with different filtering criteria. In the comparison between component level and vehicle level, the aim was to discover whether the 1% vehicle at the vehicle level was suitable for a description of representative component usage. The results show that the 1% vehicle at the vehicle level could also be used for component dimensioning in this investigation, if slight oversizing is accepted in the design. An investigation into the required number of vehicles indicates that a minimum of 10,000 vehicles is

necessary to ensure statistical robustness. This corresponds to approximately 4000 efficient vehicles with an annual mileage of 12,000 km. Additionally, a comparative analysis of driving behavior between European and American customers was conducted to clarify market-specific loads on drivetrain components.

This paper analyzes customer data to identify representative customer requirements for design. However, the number of vehicles analyzed is limited compared to the number of vehicles sold on the market. To make a more stable statement, more vehicles need to be analyzed. To trace the source and understand 1% vehicles, additional analysis should be conducted on each vehicle closest to the 1% vehicle line. It is important to understand customer driving behavior and to reconstruct driving situations. In this paper, only one vehicle type was analyzed. To ensure a general common design standard or to identify special requirements, it is necessary to analyze more vehicle types. Another important investigation that should be conducted concerns the relationship between mechanical and electric components, as their damage mechanisms differ. Therefore, the 1% vehicles may be completely different regarding electric components.

In the future, we suggest a paradigm shift towards comprehensive data collection that continuously includes all markets and models. This approach aims to enhance data-driven development by emphasizing a customer-centric philosophy. We recommend systematically adjusting drivetrain component requirements to align with customer needs. This ensures a thorough understanding of real-world usage patterns, ultimately improving the efficiency and effectiveness of drivetrain development processes.

**Author Contributions:** Investigation, M.L.; writing—original draft preparation, M.L.; writing—review and editing, F.K.-D.N., Y.Ö. and M.A.; supervision, Y.Ö. and R.H.; project administration, M.A. All authors have read and agreed to the published version of the manuscript.

**Funding:** This research received no external funding.

**Data Availability Statement:** The datasets presented in this article are not readily available because of company data security limitations.

**Acknowledgments:** This work was supported by CARIAD and Audi. We fully appreciate the effort and cooperation of Tobias Haag and Tobias Schwarzmann.

**Conflicts of Interest:** The authors declare no conflicts of interest. Mingfei Li, Fabian Kai-Dietrich Noering, Yekta Öngün and Michael Appelt are employees of Volkswagen AG. The paper reflects the views of the scientists, and not the company.

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
