# Peer review of "An Investigation of Representative Customer Load Collectives in the Development of Electric Vehicle Drivetrain Durability"

_wevj, doi:10.3390/wevj15030112_

Round 1
Reviewer 1 Report
Comments and Suggestions for Authors
Pg. 3: The author does not really define what they mean by "1%" whether it is 1% vehicle or 1% customer. In page 3 where these terms are first used, they should be properly defined and explained.
The way this reference is used is abysmal: "Special considerations for drivetrain durability include [27]." Am I supposed to go and read another paper just to complete your sentence?
Line 266: Your histogram is left skewing; therefore, your median value should be less than your mean value. However, you are indicating a mean of 11k and a median of 12k. It is not possible.
Figures 10, 11, 12, 13, 14, 15, 17, 18, 19: The y-axis lacks i) units ii) values. Please indicate these values on the graph.
Figures 8, 10, 12, 13, 15, 18: The figure legends are confusing. You are indicating numbers (10809) to represent category (performance, basic, EU, US). It is better just to indicate the category directly. Only in figure 18, it make sense to use numbers.
Line 297 to 307: It is not clear that you are talking about the 0km selection criteria in this paragraph, and it made it very confusing. I only understood that you are talking about the 0km selection criteria when I started reading the next paragraph. Please indicate clearly in the paragragh.
Line 317 to 319 and Figure 11: Your statement do not match what the figure shows, and I am not sure how figure 11 is different from figure 10? You have combined all the % categories into one?
Please improve the way you are using the reference: "To combine the collectives for different components, a common method is introduced in [13]." Now, I have to read the other paper to know the name of this common method?
Line 405 to 408: If the requirement is to "obviate" undersizing, then why are you trying to "improve" undersizing?
Figure 16: I cannot see the yellow line at all, it is invisible. Please use a more visible color.
Figure 16: The legends are confusing? What do the abbreviations represent? They have not been explained in the text either.
Compared to the number of results presented, the conclusion is very light. Please make a comprehensive evaluation to conclude.
Comments on the Quality of English LanguageN/A
Author Response
Dear Editor and Reviewers,
Thank you for your feedback on February 14. We appreciate the time and effort you put into reviewing our manuscript. Your suggestions have helped us improve our work. We value your comments and suggestions.
We have uploaded the revised manuscript file according to your instructions. We have also included a copy of the original manuscript with the changes highlighted in different colors. Please find our point-by-point response to the reviewers' comments in the attached cover letter.
We would also like to thank you for allowing us to resubmit a revised copy of the manuscript. We hope that the revised manuscript will be accepted for publication in the World Electric Vehicle Journal.
Sincerely,
Mingfei Li

Reviewer 2 Report
Comments and Suggestions for Authors
- Dear authors, the introduction to your study is particularly lacking. Please make an effort to present the background more effectively, taking into consideration robust and precise references. For example, to demonstrate how electric vehicles are surpassing conventional technology, cite studies that proves their superiority in terms of environmental performance, referencing at least one article on the life cycle assessment (LCA) of vehicles, for example:
Silvestri, L., Forcina, A., Arcese, G., & Bella, G. (2019). Environmental analysis based on life cycle assessment: An empirical investigation on the conventional and hybrid powertrain (No. 2019-24-0245). SAE Technical Paper.
and one on the well-to-wheel (WTW) analysis of fossil fuels and electricity, for example:
Yazdanie, M., Noembrini, F., Heinen, S., Espinel, A., & Boulouchos, K. (2016). Well-to-wheel costs, primary energy demand, and greenhouse gas emissions for the production and operation of conventional and alternative vehicles. Transportation Research Part D: Transport and Environment, 48, 63-84.
- In general, add at least 10 new citations to your Introduction section.
- Additionally, in the Introduction section, the authors should highlight the distinctiveness of their study compared to others. I recommend a thorough consideration of two aspects: the necessity for a new paper on this topic and the unique outcome specific to their research. The introduction should be crafted to guide readers in comprehending these two crucial aspects.
- Also, the abstract requires enhancement, explicitly articulating the issue, outlining the research objectives, specifying the methodology, and providing comprehensive details on the results and key conclusions.
- Use the same style for graphs and be sure to use axis title including unit of measurements.
- Please, in Figure 16 do not cover graphs with legend.
- The paper looks to not have any limitations. Please, highlight limitations of your work in the Conclusion Section.
Author Response

(The authors gave the same response as above.)

Reviewer 3 Report
Comments and Suggestions for Authors
Dear authors, please accept my comments as recommendations only:
1. The literature review could be extended, and the sources better analysed. As suggested in line 108, a ‘novel’ method is presented, but currently, the novelty is not convincing. You may want to provide a better comparison to the presently available methods, suggesting the knowledge gaps your research completes. With this, the problem you solve would be better presented.
2. As the requirements are an essential part (lines 81, 107, fig. 1, etc.), you may want to clarify the general requirements and the derived requirements for the sub-systems and components. The main requirements hierarchy, including functional, performance, and physical requirements, could be shown with a diagram.
3. The readers will be better introduced if a block diagram of the electric drivetrain is offered, showing the main points for data collection.
4. You may want to numerically quantify the collection data explanations (paragraph lines 114 - 143), suggesting precise numbers or ranges for the frequency, timing, volumes, etc.
5. Line 160 – you may want to clarify the described components by giving examples. Also, in this paragraph (158-177), a table with the more vulnerable components could be included, showing the factors with the most significant impact on reliability.
6. You may want to add citations to the testing and design standards.
7. The paper could be better organised. Usually, the literature review is presented initially, showing the problem and the suggested solution. Analysing the cited sources in the final parts (for example, line 388) is confusing.
Thank you for the interesting paper.
Author Response

(The authors gave the same response as above.)

Round 2
Reviewer 2 Report
Comments and Suggestions for Authors
Dear authors, the introduction to your study is particularly lacking. Please make an effort to present the background more effectively, taking into consideration robust and precise references. For example, to demonstrate how electric vehicles are surpassing conventional technology, cite studies that proves their superiority in terms of environmental performance, referencing at least one article on the life cycle assessment (LCA) of vehicles, for example:
Silvestri, L., Forcina, A., Arcese, G., & Bella, G. (2019). Environmental analysis based on life cycle assessment: An empirical investigation on the conventional and hybrid powertrain (No. 2019-24-0245). SAE Technical Paper.
and one on the well-to-wheel (WTW) analysis of fossil fuels and electricity, for example:
Yazdanie, M., Noembrini, F., Heinen, S., Espinel, A., & Boulouchos, K. (2016). Well-to-wheel costs, primary energy demand, and greenhouse gas emissions for the production and operation of conventional and alternative vehicles. Transportation Research Part D: Transport and Environment, 48, 63-84.
Reviewer 3 Report
Comments and Suggestions for Authors
The paper has been significantly improved, and my questions have been answered. I would recommend the paper to be published in the journal.